

# Improving trajectory calculations by FLEXPART 10.4+ using deep learning inspired single image superresolution

Rüdiger Brecht[1], Lucie Bakels[2], Alex Bihlo[3], and Andreas Stohl[2]

[1]Department of Mathematics, University of Bremen
[2]Department of Meteorology and Geophysics, University of Vienna, Josef-Holaubek-Platz 2
[3]Department of Mathematics and Statistics, Memorial University of Newfoundland

**Correspondence:** Rüdiger Brecht (rbrecht@uni-bremen.de)

**Abstract.** Lagrangian trajectory or particle dispersion models as well as semi-Lagrangian advection schemes require meteorological data such as wind, temperature and geopotential at the exact spatio-temporal locations of the particles that move independently from a regular grid. Traditionally, this high-resolution data has been obtained by interpolating the meteorological parameters from the gridded data of a meteorological model or reanalysis, e.g. using linear interpolation in space and
time. However, interpolation errors are a large source of error for these models. Reducing them requires meteorological input fields with high space and time resolution, which may not always be available and can cause severe data storage and transfer problems. Here, we interpret this problem as a single image superresolution task. That is, we interpret meteorological fields available at their native resolution as low-resolution images and train deep neural networks to up-scale them to higher resolution, thereby providing more accurate data for Lagrangian models. We train various versions of the state-of-the-art *Enhanced*
*Deep Residual Networks for Superresolution (EDSR)* on low-resolution ERA5 reanalysis data with the goal to up-scale these data to arbitrary spatial resolution. We show that the resulting up-scaled wind fields have root-mean-squared errors half the size of the winds obtained with linear spatial interpolation at acceptable computational inference costs. In a test setup using the Lagrangian particle dispersion model FLEXPART and reduced-resolution wind fields, we demonstrate that absolute horizontal transport deviations of calculated trajectories from "ground-truth" trajectories calculated with undegraded $0.5° \times 0.5°$ winds
are reduced by at least 49.5% (21.8%) after 48 hours relative to trajectories using linear interpolation of the wind data when training on $2° \times 2°$ to $1° \times 1°$ ($4° \times 4°$ to $2° \times 2°$) resolution data.

## 1 Introduction

Recent years have seen a considerable increase of interest in the application of machine learning to virtually all areas of the mathematical sciences, with meteorology being no exception. Machine learning, and specifically deep learning, which is
concerned with training deep artificial neural networks, holds great promise for problems for which vast amounts of data are available. This is the case for meteorology, where a dense network of observational instruments, such as satellites, rawinsondes, and ground-based stations in conjunction with sophisticated numerical weather prediction and reanalysis models generate a large store of available data, which are a prerequisite for training deep neural networks. Breakthroughs in the availability of affordable graphics processing units and substantial improvements in training algorithms for deep neural networks have equally



contributed to making deep learning a promising new tool for applications in computer vision Krizhevsky et al. (2012), speech generation Oord et al. (2016), text translation and generation Vaswani et al. (2017), and reinforcement learning Silver et al. (2017), just to name a few. Applications of deep learning to meteorology so far include weather nowcasting Shi et al. (2015), weather forecasting Rasp et al. (2020); Weyn et al. (2019), ensemble forecasting Bihlo (2021); Brecht and Bihlo (2022); Scher and Messori (2018), subgrid-scale parameterization Gentine et al. (2018), and downscaling Mouatadid et al. (2017).

In recent years image super resolution by neural networks has made considerable progress. The main application is to scale a low resolution image to an image with a higher resolution, which is referred to as *single image superresolution* (SISR), although similar techniques are also used to up-scale both the spatial resolution and the frame rates for videos as well. Before the advent of efficiently trainable convolutional neural networks, the superresolution problem for images was solved using interpolation based methods, see e.g. Li and Orchard (2001), which is surprisingly still the state of the art for Lagrangian models.

Semi-Lagrangian advection schemes in numerical weather prediction models rely on simple interpolation methods for the wind components  Durran (2010). For instance, the semi-Lagrangian scheme in the Integrated Forecast System model of the European Centre for Medium Range Weather Forecasts (ECMWF) uses a linear interpolation scheme. In trajectory models and Lagrangian particle dispersion models, similarly simple interpolation methods are used. Higher-order interpolation schemes such as bicubic interpolation can reduce the wind component interpolation errors compared to linear interpolation Stohl et al.

(1995). However, error reductions for higher-order schemes are less than 30%, while computational costs increase by about an order of magnitude Stohl et al. (1995). Therefore, many trajectory and Lagrangian particle dispersion models still use linear interpolation, e.g., FLEXPART Pisso et al. (2019), LAGRANTO Sprenger and Wernli (2015), or MPTRAC Hoffmann et al. (2021)

     The purpose of this paper is to implement variable-scale superresolution based on deep convolutional neural networks to
showcase their potential for Lagrangian models. Here, we make use of the self-similarity of meteorological fields, such that a neural network can be repeatedly applied to interpolate a velocity field to higher resolutions. The paper's further organization is as follows. In Section 2 we review some of the common architectures used in computer vision to train SISR models. Here, we choose the Enhanced Deep Residual Network for Single Image Super-Resolution (EDSR), which is near state-of-the-art for SISR in computer vision, and which is straightforward to use for meteorological fields. Section 3 describes the numerical setup
and the data being used in this work. In Section 4 we present the results of our study, illustrating substantial improvements of both the quality of up-scaled wind fields using the EDSR model in comparison to standard linear interpolation, as well as of trajectory calculations using the Lagrangian particle dispersion model FLEXPART Pisso et al. (2019). A summary of this paper and thoughts for future research can be found in Section 5.

## 2  Related work

SISR is a topic of substantial interest in computer vision, with applications in computational photography, surveillance, medical imaging and remote sensing Chen et al. (2022). A variety of architectures have been proposed in this regard, essentially all of which use a convolutional neural network architecture, following the seminal contribution Krizhevsky et al. (2012) which





kindled the explosive interest in modern deep learning. Among these SISR architectures, some important milestones are Super-Resolution Convolutional Neural Network (SRCNN) Dong et al. (2014), a standard convolutional neural network, Very Deep

Super Resolution (VDSR) Kim et al. (2016), a convolutional neural network based on the popular Visual Geometry Group (VGG) architecture (a standard deep convolutional neural network architecture with multiple layers), Super Resolution Generative Adversarial Network (SRGAN) Ledig et al. (2017), a generative adversarial network, and EDSR Lim et al. (2017), based on a convolutional residual network architecture. For a recent review on SISR providing an overview over the aforementioned architectures and others, the reader may wish to consult Yang et al. (2019).

While deep learning has been used extensively over the past several years in meteorology for a variety of use cases, including weather prediction Rasp et al. (2020); Weyn et al. (2019), ensemble prediction Bihlo (2021); Brecht and Bihlo (2022); Scher and Messori (2021), nowcasting Bihlo (2019); Shi et al. (2015), downscaling Mouatadid et al. (2017); Sha et al. (2020), and subgrid-scale parameterization Gentine et al. (2018), there have only been a few applications of deep learning to meteorological interpolation that go beyond downscaling. This is surprising, as many tasks in numerical meteorology routinely involve

interpolation, such as the time-stepping in numerical models using the semi-Lagrangian method requiring trajectory origin interpolation Durran (2010), or Lagrangian particle models Stohl et al. (2005).

## 3   Methods

For the training and evaluation we note that meteorological fields are characterized by self-similarity over a variety of spatio-temporal scales. This makes it possible to train the neural network model to increase the resolution from a down-sampled

velocity field to a higher resolution and then apply the model repeatedly to obtain even higher resolutions.

Below we introduce the data used to train the neural network, describe the details of the neural network model and how we train the model. Moreover, we explain how we used the interpolated fields to run a simulation with FLEXPART.

### 3.1   Training data

To train our neural networks, we use data from the ECMWF ERA5 reanalysis Hersbach et al. (2020). The data are available at

an hourly global spatial resolution of $0.5° \times 0.5°$ in latitude–longitude coordinates and a total of 138 vertical levels. We use a total of 296 hours (from January 1 to January 12, 2000) for training and test our model for 24 hours in each season (on January 15, April 15, July 15, October 15, 2000). While this may seem like comparatively little data, as we train a two-dimensional model on each horizontal layer, each hourly data point corresponds to a total of 138 layers, yielding a total of roughly 15000 sample fields.

The low-resolution data is obtained from the high-resolution ERA5 data by simply sampling every 1st, 2nd and 4th degree. In this work we focus solely on the spatial uscaling problem. The temporal upscaling problem will be considered elsewhere; see further discussions in the conclusions. We also only interpolate the horizontal wind components, and interpolate only horizontally.



### 3.2 Neural network architecture

We use the EDSR architecture Lim et al. (2017) with additional channel attention. The main building block of this architecture is a simplified version of a standard convolutional residual network block without batch normalization (Fig. 1b). This residual block consists of two convolutional layers, each of which uses a filter size of 3 in the present work, with the first convolutional layer being followed with a standard rectified linear unit activation function. After the second convolutional layer, a scaling of the output feature maps is performed, where we use the same constant residual scaling factor of 0.1 as proposed in the

original work Lim et al. (2017). The final operation of each residual block is given via channel attention (Fig. 1c). The overall architecture of this channel attention module follows Choi et al. (2020). The purpose of attention mechanisms in a convolutional neural network is to enable it to focus on the most important regions of the neural network's perceptive field. Channel attention re-weights each respective feature map from a convolutional layer following a learn-able scale-transformation. The last building block of our architecture is the upsampling module (Fig. 1d). This module consists of a convolutional layer with a total of 64

$\times$ `upscale_factor`$^2$ feature maps, followed by a depth-to-space transformation called PixelShuffle Shi et al. (2016) which re-distributes feature maps into an `upscale_factor`-times larger spatial field. In this work, `upscale_factor` $= 2$.

The main residual network blocks are repeated 8 times, with an extra skip connection being added before the first convolutional layer in the network and before the upsample module. The upsampled image passes all convolutions in our architecture, with the exception of the convolutional layer in the upsample module using a total of 64 filters.

We have experimented with a variety of other architectures, including a conditional convolutional generative adversarial network called `pix2pix` Isola et al. (2017) and the super-resolution GAN Ledig et al. (2017), but have found the EDSR network giving the most impressive results with the greatest ease of training and setup. Hence we exclusively report the results from the EDSR model below.

### 3.3 Neural network training

For each velocity component $u$ and $v$ we train a separate neural network to interpolate a field from degraded $2° \times 2°$ resolution data to $1° \times 1°$ resolution data, we call this neural network `model2`. In a second experiment, another set of neural networks (`model4`) is trained to interpolate the respective velocity fields from degraded $4° \times 4°$ resolution data to $2° \times 2°$ resolution data, with the goal to then apply the trained model again to obtain the fields at the $0.5° \times 0.5°$ resolution. Moreover, for testing purposes we train a neural network (`model1`) to interpolate $1° \times 1°$ to $0.5° \times 0.5°$ resolution.

One neural network is trained for the lower atmospheric levels (until level 50) and another for the higher ones (level 51 to 138), because the lower and higher level fields have a rather different structure owing to the vertical stratification of the atmosphere. Each neural network is trained on the data of 294 hourly wind fields with 50 or 88 vertical levels, this results in 14700 or 25578 training samples, respectively.





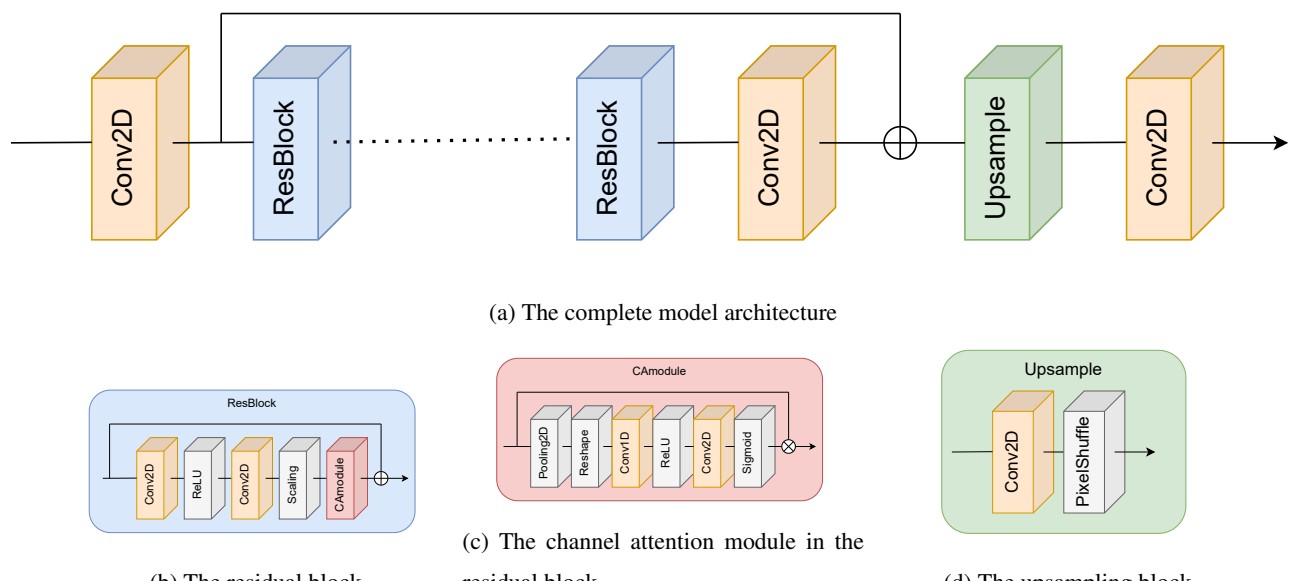

(a) The complete model architecture

(b) The residual block

(c) The channel attention module in the residual block.

(d) The upsampling block.

**Figure 1.** The overall layout of the neural network model is presented which consists of an upsampling block and residual blocks, which again contain a channel attention module. Here, $\oplus$ means adding the layers and $\otimes$ multiplying them.

The model was implemented using TensorFlow 2.8 and will be after publication publicly available on Github[1]. We trained the model on a dual NVIDIA RTX 8000 machine, with each training step taking approximately 100 ms for the $u$ and $v$ field. Total training took roughly 2.5 hours for each field.

### 3.4 Interpolation error metrics

In the following we report the results of our study using the root mean square error (RMSE) and the structural similarity index measure (SSIM) as performance measures. The RMSE is defined as

$$\mathrm{RMSE}_z = \sqrt{\overline{(z_{\mathrm{interpolated}} - z_{\mathrm{reference}})^2}}, \tag{1}$$

where in the following $z \in \{u, v\}$, with the bar denoting spatial averaging. The reference solution is given by the original $0.5° \times 0.5°$ ERA5 data, assumed to represent the ground truth. The smaller the RMSE, the better the interpolated results coincide with the original reference solution.

The SSIM is a measure of the perceived similarity of two images and is defined as

$$\mathrm{SSIM}(x, y) = \frac{(2\mu_x\mu_y + C_1)(2\sigma_{xy} + C_2)}{(\mu_x^2 + \mu_y^2 + C_1)(\sigma_x^2 + \sigma_y^2 + C_2)}. \tag{2}$$

---

[1]https://github.com/RudigerBrecht/Improving-trajectory-calculations-using-SISR



with $\mu_x$ and $\mu_y$ denoting the means of the two images $x$ and $y$ (computed with an $11 \times 11$ Gaussian filter of width 1.5), $\sigma_x$ and $\sigma_y$ denoting their standard deviations and $\sigma_{xy}$ being their co-variance. The constants $C_1$ and $C_2$ are defined as $C_1 = (K_1 L)^2$ and $C_2 = (K_2 L)^2$, respectively, with $K_1 = 0.01$ and $K_2 = 0.03$ and $L = 1$. The closer the SSIM value is to 1, the more similar the two images are. See Wang et al. (2004) for further details. In the following $x = z_{\text{interpolated}}$ and $y = z_{\text{reference}}$, with each of them being interpreted as a gray-scale image.

### 3.5 Trajectory calculations

To test the impact of the neural network interpolated wind fields on trajectory calculations, we used the Lagrangian particle dispersion model FLEXPART Pisso et al. (2019); Stohl et al. (2005). The calculations were based on version 10.4 of FLEXPART, which has been modified to interpolate directly the ECMWF model-level data, instead of first transforming it to terrain-following coordinates. We switched off all turbulence and convection parameterizations and used FLEXPART as a simple trajectory model. Ideally, the neural network interpolation should be implemented directly in FLEXPART. However, as the neural network and FLEXPART run on different computing architectures (Graphics vs. Central Processing Unit), this is outside of the scope of this exploratory study. Instead, we replaced the gridded ERA5 wind data with the gridded up-sampled testing data produced by the neural network. This does not make full use of the neural network capabilities, as we ingest these data at a fixed resolution of $0.5° \times 0.5°$ latitude/longitude and use linear interpolation of the wind data to the exact particle position, while in principle the neural network could also determine the wind components almost exactly at the particle positions (upon repeatedly using the trained SISR model to increase the resolution high enough to obtain the wind values at the respective particle positions). FLEXPART also needs other data than the wind data, for which we use linear interpolation of the ERA5 data. For temporal interpolation, we also used linear interpolation, as is standard in FLEXPART.

We started multiple simulations with 10 million trajectories on a global regular grid with 138 vertical levels and traced the particles for 48 hours. This simulation was repeated in each season for the following cases:

- the original ERA5 data at $0.5° \times 0.5°$ resolution, serving as the *ground truth* reference case;

- a data set, for which the winds were interpolated from degraded $1° \times 1°$ and from $2° \times 2°$ resolution data, using linear interpolation (the *linear interpolation* case);

- a data set, for which the winds were interpolated from degraded $1° \times 1°$ (using the neural network `model2` trained to interpolate $2° \times 2°$ to $1° \times 1°$) and from $2° \times 2°$ resolution data (using the neural network `model4` trained to interpolate $4° \times 4°$ to $2° \times 2°$), and then interpolated to the particle position using linear interpolation in FLEXPART (the *neural network interpolation* case).

### 3.6 Trajectory error metrics

As in previous studies Kuo et al. (1985); Stohl et al. (1995), we compared the trajectory positions for trajectories calculated with the interpolated data to those calculated with the reference data set, using the Absolute Horizontal Transport Deviation





(AHTD), defined as:

$$\text{AHTD} = \frac{1}{N} \sum_{n=1}^{N} D[(X_n, Y_n), (x_n, y_n)] \tag{3}$$

where $N$ is the total number of trajectories, $D[(X_n, Y_n), (x_n, y_n)]$ is the great circle distance of trajectory points with longi-

tude/latitude coordinates $(X_n, Y_n)$ for the reference trajectories and $(x_n, y_n)$ for the trajectories using interpolated winds, for
trajectory pair $n$ starting at the same point. AHTD values are evaluated hourly along the trajectories, up to 48 hours.

## 4   Results

In this section we first show that the interpolation using the neural network gives better results compared to the linear interpola-
tion. Then, we demonstrate that the trajectories computed with FLEXPART using the interpolated fields of the neural network

are more accurate compared to linear interpolation.

### 4.1   Interpolation

We demonstrate the self-similarity of the spatial scales by interpolating the fields multiple times using the same model trained
to up-scale the wind fields from lower resolutions. For the interpolation we use linear and neural network interpolation. First,
we compare the fields which are interpolated from $1° \times 1°$ to $0.5° \times 0.5°$ resolution data. Then, we demonstrate that the neural

network interpolation can be used multiple times to generate arbitrary resolution.

In Fig. 2 we show the interpolation results for three different neural networks and for the linear interpolation. We see that
each neural network has better metrics (i.e., lower RMSE and higher SSIM values) than the corresponding linear interpolation.
This is true both for the resolution the neural network has been trained for, as well as higher resolutions.

### 4.1.1   One time up-scaling

We consider the neural network `model2` (trained to interpolate $2° \times 2°$ to $1° \times 1°$ resolution data). For the evaluation we
interpolate a field from degraded $1° \times 1°$ resolution data to $0.5° \times 0.5°$ resolution data. Notice that we have trained the model
separately for levels 0–50 and 51–138, and evaluate the correspondingly trained model.

In Fig. 3 and Fig. 4 we show the RMSE for the interpolated field for level 10 as an example and observe that the neural
network interpolation has overall a lower RMSE and is closer to the reference velocity field. With the linear interpolation,

large errors occur especially near fronts or shear zones, and these errors are substantially reduced by the neural network
interpolation. This also means that especially the largest interpolation errors are avoided by the neural network, compared to
the linear interpolation (see Fig. 5). More example figures for the different months (January, April, July and October) can be
seen on the git repository. We note that the results are similar, which can be seen in Table 1, where we computed the RMSE
and SSIM for a whole day in the months January, April, July and October and all levels. The neural network interpolation has

less than half the RMSE of the linear interpolation and achieves a higher SSIM value.




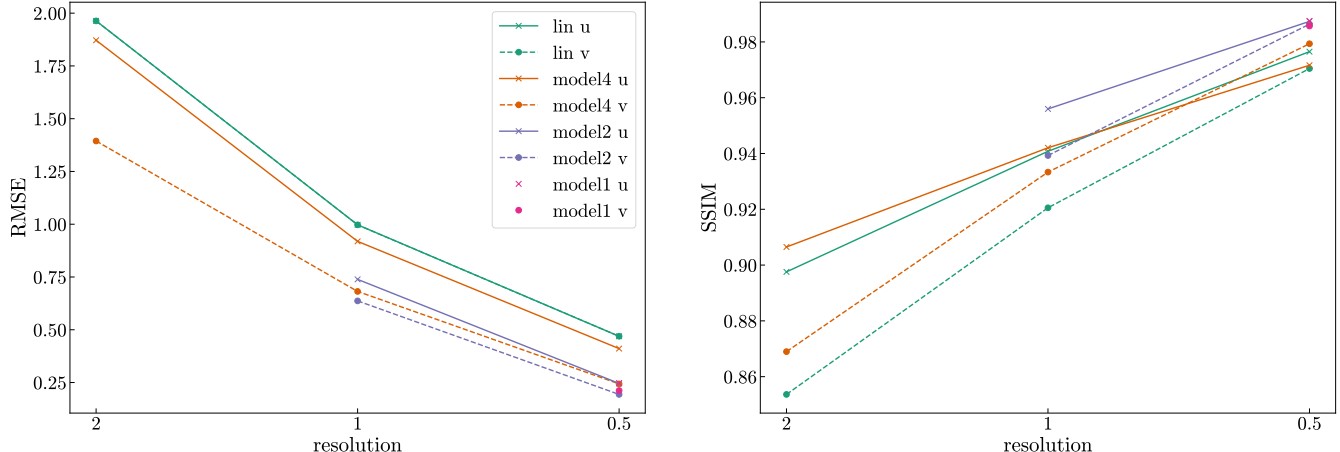

**Figure 2.** RMSE and mean SSIM of the validation data set (14th January 2000) for linear and neural network interpolation evaluated at different resolutions. Here, we do not interpolate the fields multiple times (as explained in section 4.1.2) but rather once for each resolution starting from the resolution the model is trained on. The solid lines are computed for the $u$ velocity and the dashed lines for the $v$ velocity.

| RMSE ↓ | Linear | | Neural Network | | SSIM ↑ | Linear | | Neural Network | |
|---|---|---|---|---|---|---|---|---|---|
| | u | v | u | v | | u | v | u | v |
| January | 0.469 | 0.398 | 0.214 | 0.181 | January | 0.976 | 0.970 | 0.988 | 0.987 |
| April | 0.393 | 0.36 | 0.206 | 0.182 | April | 0.976 | 0.968 | 0.988 | 0.986 |
| July | 0.447 | 0.444 | 0.222 | 0.193 | July | 0.975 | 0.968 | 0.988 | 0.986 |
| October | 0.589 | 0.532 | 0.216 | 0.191 | October | 0.975 | 0.969 | 0.988 | 0.987 |

**Table 1.** RMSE and mean SSIM of the validation data set for linear and neural network interpolation using `model2`. Considering the RMSE, the neural network interpolation is at least 49% more accurate compared to the linear interpolation.

Moreover, we note that the interpolation time for one field for a given level and time for the linear interpolation is about 0.002 s while the same interpolation using the neural network takes about 0.02 s.

### 4.1.2 Multiple time up-scaling

To demonstrate that the neural network can be used to interpolate a field to arbitrary resolution we consider the neural network 195 `model4`( trained to interpolate $4° \times 4°$ to $2° \times 2°$ resolution data). We evaluate the network to interpolate $2° \times 2°$ to $1° \times 1°$ and evaluate it another time to interpolate $1° \times 1°$ to $0.5° \times 0.5°$ resolution data. In Fig. 6 and Fig. 7 we show the RMSE of an interpolated field in January at level 10 (around 220m) and compare it to the RMSE of the linear interpolation.





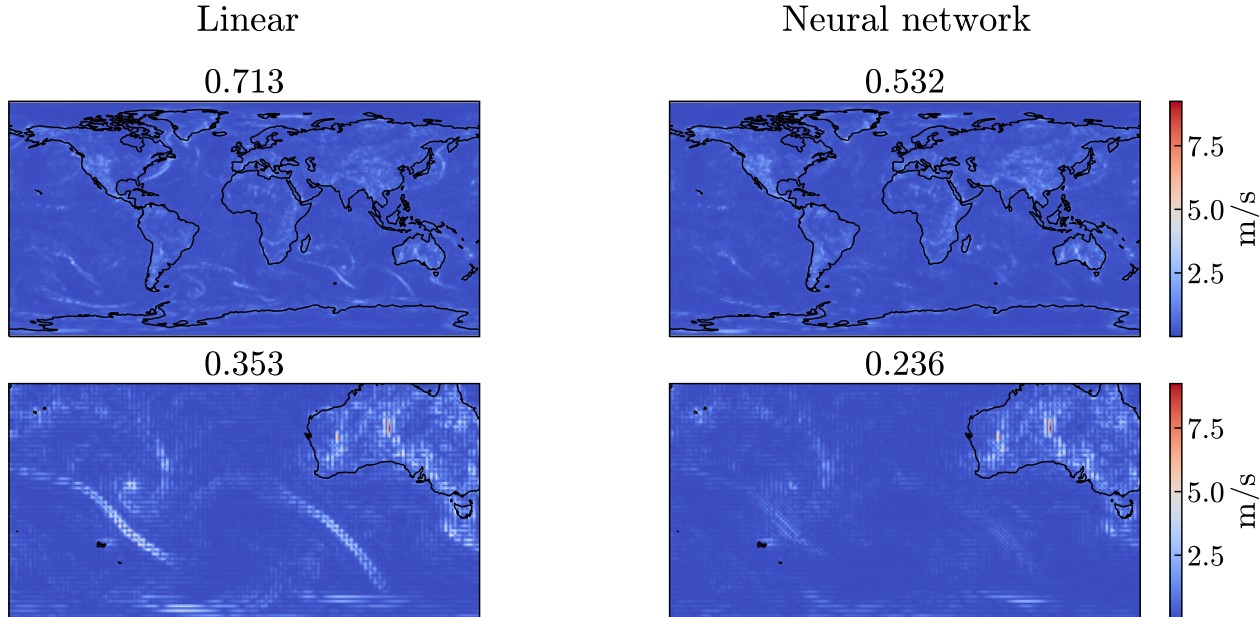

**Figure 3.** Differences of $\mathbf{v} = (u, v)$ field on 14th January 2000 at 10:00 UTC for level 10 (around 220m) when compared to the ground truth ERA5 data, for linear interpolation (left panels) and for the interpolation by the `model2` neural network (right panels). The data is interpolated from degraded $1° \times 1°$ resolution data. The bottom row shows a blown-up section of each field. The title of each plot shows the RMSE of the region.

For each method the RMSE is higher than before, since we start with a lower resolution and less information. Nevertheless, the RMSE of the neural network interpolation is again lower compared to the linear interpolation, albeit the relative error reduction is smaller than with one-time up-scaling. This also holds for other samples which we omitted showing here. When evaluating the RMSE for a day in January, April, July and October for all levels (Table 2), we observe that the neural network interpolation again achieves better results. Here, we are limited to the resolution of the reference data. Thus, we can only demonstrate the interpolation for two times. Even coarser data does not have enough small scales represented, such that it is not meaningful to train the network on even coarser data and upscale the data more often.



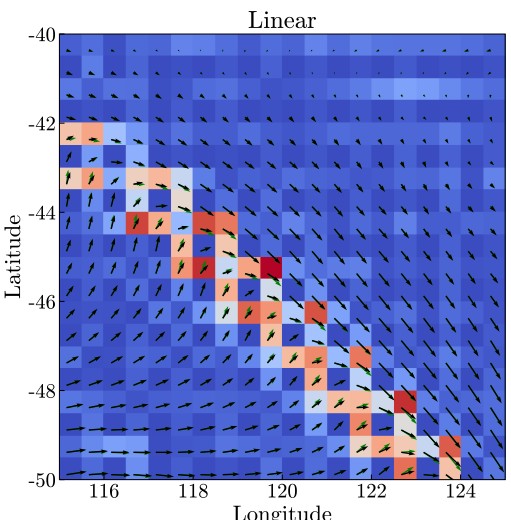
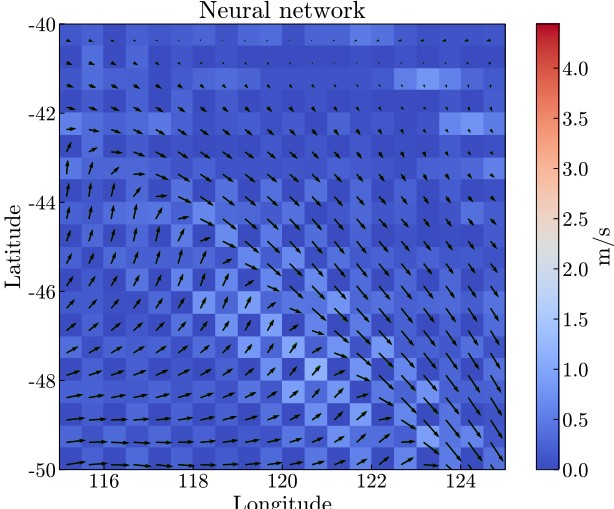

**Figure 4.** Comparison of a section of the $\mathbf{v} = (u, v)$ vector field at level 10 using linear interpolation and the interpolation by the neural network `model2` (black arrows). The date of the field is 14th January 2000 at 10:00 UTC. The data is interpolated from degraded $1° \times 1°$ resolution data. The arrows in green show the reference vector field and the color-bar shows the RMSE. Notice that the checkerboard structures occurring with the linear interpolation are caused by the fact that winds in every fourth grid cell do not have to be interpolated and are therefore error-free.

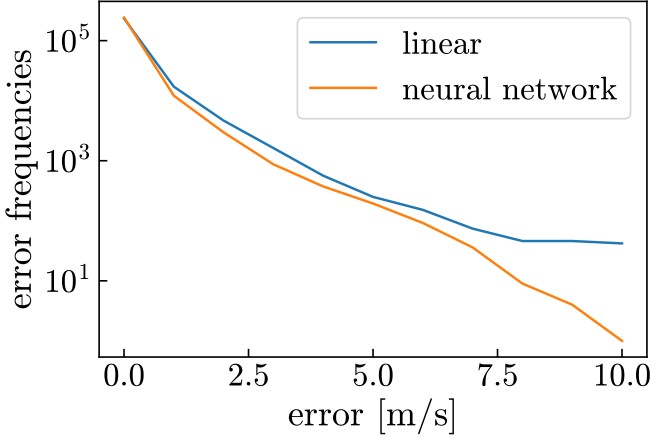

**Figure 5.** Comparison of the error frequencies for linear and neural network interpolation (`model2`) for the same field as in Fig. 3 (January 2000 at 10:00 UTC). Here, we split the error frequencies into 10 bins of different intensities.





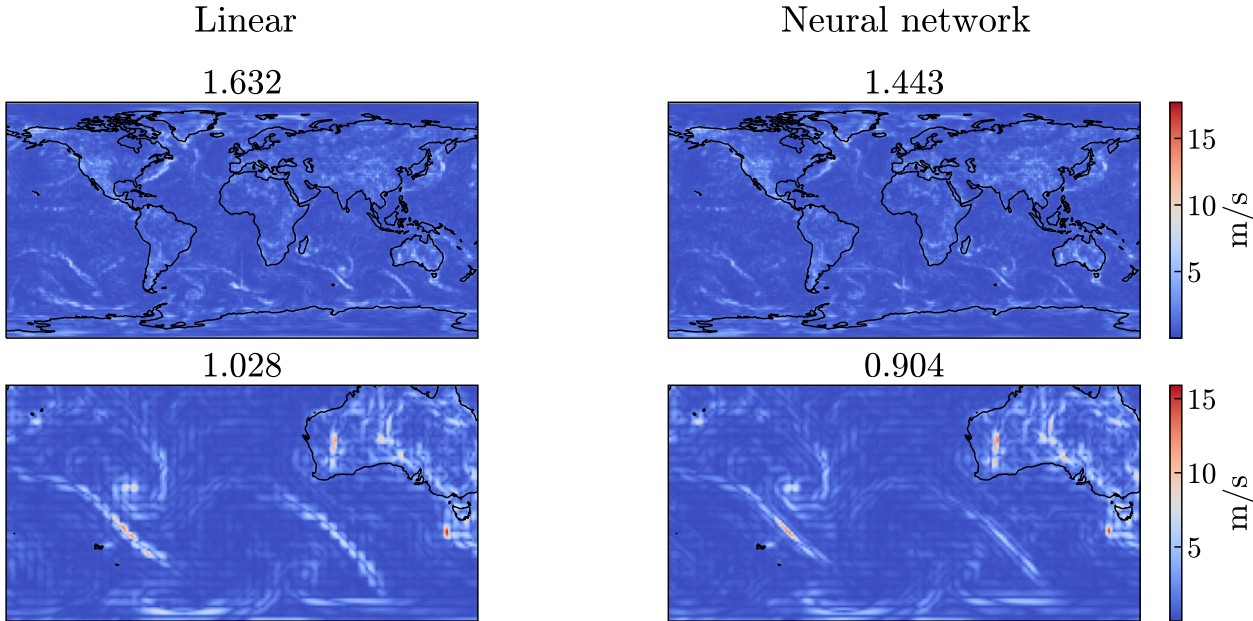

**Figure 6.** RMSE of $\mathbf{v} = (u, v)$ field at level 10 when compared to the ground truth ERA5 data. The date of the field is 14th January 2000 at 10:00 UTC. Linear interpolation (left panels) and the interpolation by the neural network `model4` (right panels). The data is interpolated from degraded $2° \times 2°$ resolution data. The bottom row shows a section of each field. The title of each plot shows the RMSE.

| RMSE ↓ | Linear | | Neural Network | | SSIM ↑ | Linear | | Neural Network | |
|---|---|---|---|---|---|---|---|---|---|
| | u | v | u | v | | u | v | u | v |
| January | 1.107 | 0.938 | 0.787 | 0.708 | January | 0.864 | 0.824 | 0.892 | 0.849 |
| April | 0.96 | 0.863 | 0.777 | 0.677 | April | 0.860 | 0.808 | 0.882 | 0.837 |
| July | 1.095 | 1.031 | 0.84 | 0.77 | July | 0.856 | 0.809 | 0.881 | 0.835 |
| October | 1.289 | 1.155 | 0.796 | 0.744 | October | 0.862 | 0.820 | 0.890 | 0.845 |

**Table 2.** RMSE and mean SSIM of the validation data set for linear and neural network interpolation, using `model4`. Considering the RMSE, the neural network interpolation is at least 19% more accurate compared to the linear interpolation.





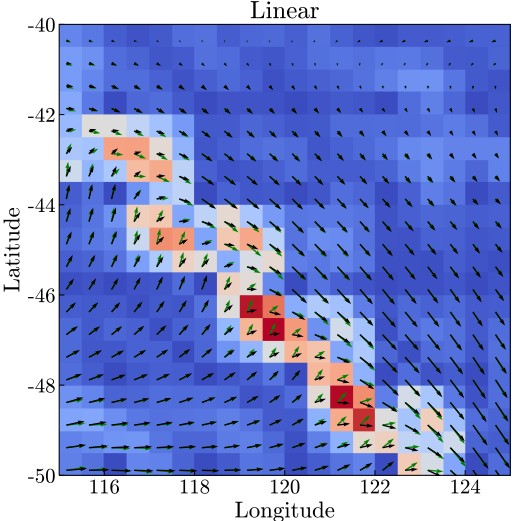
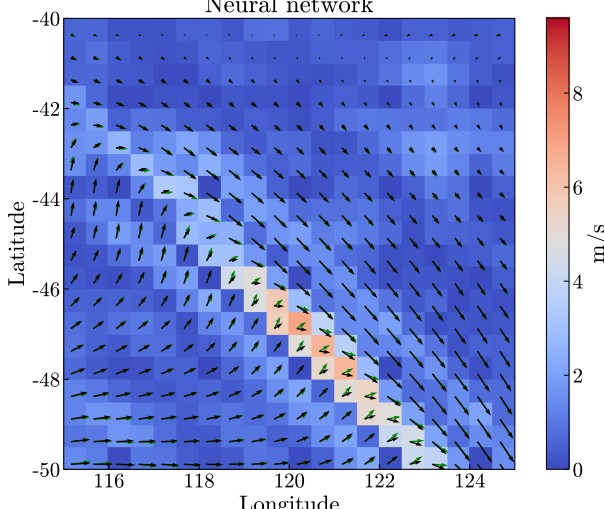

**Figure 7.** Comparison of a section of the $\mathbf{v} = (u, v)$ vector field at level 10 using linear interpolation and the interpolation by the neural network `model4` (black arrows). The date of the field is 14th January 2000 at 10:00 UTC. The data is interpolated from degraded $2° \times 2°$ resolution data. The arrows in green show the reference vector field and the color-bar shows the RMSE.

## 4.2 Trajectory accuracy

We have shown that neural network interpolated wind velocity fields are more similar to the original $0.5° \times 0.5°$ resolution data than their linearly interpolated equivalents. It is thus likely that trajectories calculated based on the neural network inter-

polated fields are also more accurate than those based on wind fields based on linear interpolation. However, trajectories are not always equally sensitive to wind interpolation errors, and it is therefore important to show that the individual trajectories that are advanced using neural network interpolated wind fields are indeed more similar to trajectories that are advanced using the original wind fields.

Fig. 8 shows that this is indeed the case. Both the average horizontal transport deviation from the original *ground truth* trajectories as well as its standard deviation are smaller for the neural network as compared to the linear interpolation. The absolute deviations after 48 hours are on average $\sim 53.5\%$ ($1° \times 1°$ resolution) and $\sim 29.4\%$ ($2° \times 2°$ resolution) smaller for all seasons when using the neural network. Moreover, the standard deviation of the neural network is consistently smaller, no matter the season (on average $\sim 36.1\%, \sim 17.9\%$ smaller for the $1° \times 1°$ and $2° \times 2°$ resolution, respectively). The improvement of the

neural network over the linear interpolation is smaller with multiple time up-scaling from $2° \times 2°$ resolution than with one time up-scaling from $1° \times 1°$ resolution, in agreement with the smaller wind interpolation errors in this case (see Fig. 2). The reduced standard deviation we see in the neural network interpolated trajectories as compared to the linear interpolated ones, directly corresponds to the lower frequency of extreme deviations found in the neural network interpolated wind fields as com-



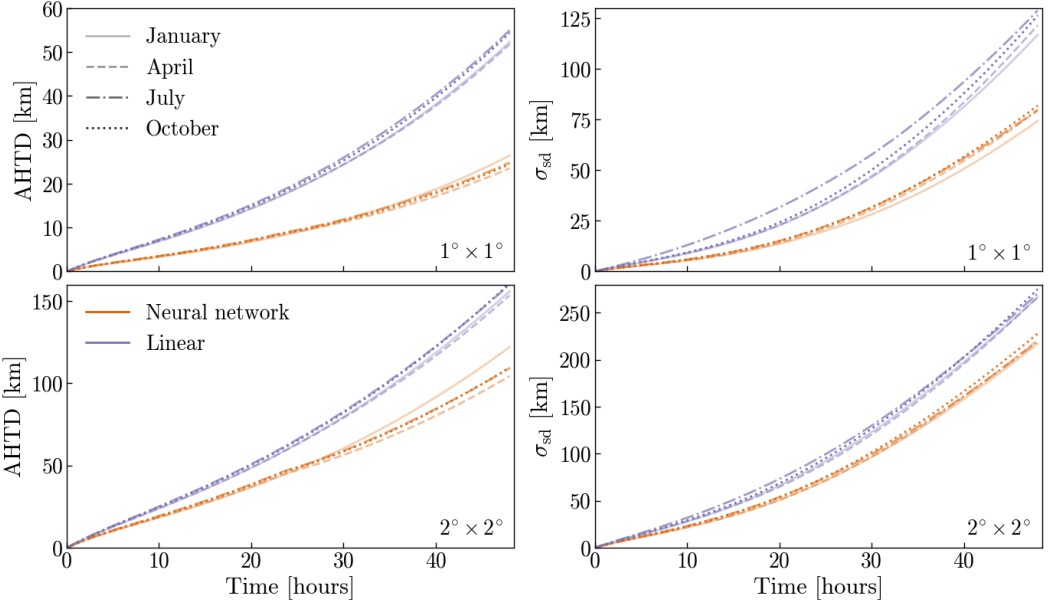

**Figure 8.** Absolute horizontal transport deviations (Eq. (3)) and standard deviations of 10 million particles advanced with FLEXPART, using two different degraded resolution data (as described in section 4.1) and two interpolation methods, as compared to the same particles advanced using the original full resolution data. The top panels show the results for the degraded $1° \times 1°$ resolution data (neural network interpolation using model2), and the bottom panels those of the degraded $2° \times 2°$ resolution data (neural network interpolation using model4). Orange lines show the AHTD (left panels) and standard deviation (right panels) of particles advanced using the neural network, and purple lines show these for the linearly interpolated data. Results for different seasons are shown with different line styles.

pared to the linear interpolated ones (see Fig. 5). Thus, trajectories using the neural network interpolation are not only more
accurate on average than trajectories using linear interpolation but large trajectory errors are avoided more efficiently as well. This is important for avoiding misinterpretation when trajectories are used to interpret source-receptor relationships e.g. for air pollutants or greenhouse gases.

We have also checked how well quasi-conserved meteorological properties such as potential vorticity and potential temperature are conserved along the trajectories. This showed generally small differences between the different trajectory data sets.
However, the trajectories based on neural network interpolated wind data had slightly better tracer conservation than the ones based on linear interpolation, confirming that these trajectories are indeed more accurate.

Notice that we have not changed vertical and time interpolation of the winds and that we have not at all changed the interpolation of the vertical wind. Furthermore, we have not made full use even of the neural network horizontal interpolation of the horizontal winds, as interpolation below $0.5° \times 0.5°$ resolution was still done using linear interpolation. We therefore consider
substantial further error reductions possible, if neural network interpolation both in space and time is fully implemented di-



rectly in the trajectory model. This also suggests that semi-Lagrangian advection schemes could be made much more accurate with neural network interpolation.

## 5 Conclusions

In this paper we have considered the problem of increasing the spatial resolution of meteorological fields using techniques
of machine learning, namely using methods originally proposed for the problem of single image superresolution. Higher-resolution meteorological fields are relevant for a variety of meteorological and engineering applications, such as particle dispersion modeling, semi-Lagrangian advection schemes, down-scaling, and weather nowcasting, just to name a few.

What sets the present work apart from a pure computer vision problem is that meteorological fields are characterized by self-similarity over a variety of spatio-temporal scales. This gives rise to the possibility of training a neural network to learn
to increase the resolution from a down-sampled meteorological field to the original native resolution of that field, and then to repeatedly apply the same model to further increase the resolution of that field beyond the native resolution. We have shown in this paper that this is indeed possible. Wind interpolation errors are at least 49% and 19% smaller than errors using linear interpolation, with one time up-scaling, and with multiple time up-scaling, respectively. Here, we note that the multiple time up-scaling has a lower improvement than the one time up-scaling because we use different neural networks based on
the available resolution. This means that the neural network trained on lower resolution data has less information and is less accurate than the neural network trained on the higher resolution data, see Fig. 2. We have also shown that corresponding absolute horizontal transport deviations for trajectories calculated using these wind fields are 52% (from degraded $1° \times 1°$ resolution data) and 24% (from degraded $2° \times 2°$ resolution data) smaller than with winds based on linear interpolation. This is a substantial reduction, given that we have not changed vertical and time interpolation and that we have not at all changed
the interpolation of the vertical wind. Furthermore, we have not even made full use of the neural network interpolation, as interpolation below $0.5° \times 0.5°$ resolution was still done using linear interpolation.

While in the present work we have exclusively focused on the spatial interpolation improvement problem, similar techniques as presented here are applicable to the temporal interpolation case as well. Here, the problem can be interpreted as increasing the frame rate in a given video clip, with the native resolution given by the temporal resolution as made available by numerical
weather prediction centres. We are presently working on this problem as well, and the results will be presented in future work. Subsequently, spatial and temporal resolution improvements can be combined to provide a seamless way to increase the overall resolution of meteorological fields for a variety of spatio-temporal interpolation problems.

Lastly, we should like to stress that meteorological fields are quite different from conventional photographic images as typically considered for superresolution tasks. Namely, meteorological fields follow largely a well-defined system of partial
differential equations, which we have not considered when increasing the spatial resolution of the given datasets. This means that potentially important meteorological constraints such as energy, mass and potential vorticity conservation may be violated by obtained up-scaled datasets, as is also the case for other interpolation methods. Incorporating these meteorological constraints would be critical if these fields would be used in conjunction with numerical solvers, and correspondingly the pro-



posed methodology would have to be modified to account for these constraints. This will constitute an important area of future

research, with a potential avenue being provided through so-called physics-informed neural networks. See e.g. Raissi et al. (2019) and Bihlo and Popovych (2022) for an application of this methodology to solving the shallow-water equations on the sphere. Physics-informed neural networks allow one to take into account both data and the differential equations underlying these data, which would enable one to train a neural network based interpolation method that is also consistent with the governing equations of hydro-thermodynamics. Including consistency with these differential equations will be another potential

avenue of research in the near future.

*Code and data availability.* The code to train the neural network and data to reproduce the plots will be available on Github: https://github.com/RudigerBrecht/Improving-trajectory-calculations-using-SISR.

*Author contributions.* All authors contributed equally to the conceptualization, methodology and writing of the manuscript. RB and LB carried out the numerical simulations and code development.

*Competing interests.* The authors declare that they have no conflict of interest.

*Acknowledgements.* This research was undertaken in part thanks to funding from the Canada Research Chairs program, the InnovateNL LeverageR&D program, the NSERC Discovery Grant program and the NSERC RTI Grant program. The study was also supported by the Dr. Gottfried and Dr. Vera Weiss Science Foundation and the Austrian Science Fund in the framework of the project P 34170-N, "A demonstration of a Lagrangian re-analysis (LARA)". Moreover, this project is funded by the Deutsche Forschungsgemeinschaft (DFG, German Research

Foundation) – Project-ID 274762653 – TRR 181.



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
