# Peer review of "Improving trajectory calculations by FLEXPART 10.4+ using single image superresolution"

_EGUsphere, 2022_

## Author Response (AR1)

**Summary**

We thank the two reviewers for providing valuable feedback on the first version of our submitted paper. We have done our best to take into account all remarks raised. In the following we give a detailed list of all the changes made in response to the points raised by the reviewers.

Thank you once more for your help in improving our paper.

**Reviewer 1**

**Major items**

1. Currently, the authors sub-sample a higher resolution model field to obtain the coarse resolution wind field. This approach is in my impression inconsistent with what a coarser-resolution model would provide. A coarser-resolution model would provide an average of what is represented by several grid points of a finer-scale model. I think the sub-sampling makes it harder for the linear interpolation to provide good results. That sub-sampling approach is also the origin of the checkerboard patterns apparent in e.g. Fig. 4. I recommend to redo the analysis by averaging the model fields rather than sub-sampling.

➔ The motivation for our choice of degrading data by leaving certain grid points entirely unchanged, is twofold:

1. This approach has been used often in the past when studying wind interpolation errors for trajectory models (e.g., Kuo et al., 1985; Stohl et al., 1995). It is appropriate to be consistent with such past approaches.

2. We agree that it would also be interesting to see how higher-resolution data could be reconstructed from lower-resolution data. This would then be more equivalent to downscaling approaches in weather prediction. However, for comparing the skill of different interpolation methods, this is not ideal. At the points where data are available at both high and low resolution, these data would be different in each case. Reconstruction of points in between would then not only reflect differences in the skill of interpolation but also the data differences at the points from where the interpolation is done. This would thus

not allow a "clean" evaluation of different interpolation methods, mixing the effects of interpolation and grid-cell averaging for the coarse-resolution data points.

2. Is it correct that the neural network works on a 3x3 grid point stencil on a lat-lon grid? It seems that the footprint of the interpolation operator would thus see very different area sizes near the pole than near the equator. How is this affecting the training and application of the neural network?

➔ It is correct that some parts of the neural network work on 3x3 grid point stencils. However, the network architecture is more complicated than considering just 3x3 stencils working on the lat-lon grid. Ideally the nonlinear nature of the neural network learns how to cope with the different area sizes. It would be very interesting to analyze the effect of the area, but this is not the scope of the present work.

3. Some sections are poorly written and therefore hardly comprehensible (for example section 3.2). Since the authors introduce new methods into the field of atmospheric science, I recommend to make an extra effort to clearly define all terminology (e.g., "channel attention" is never defined). However, I do not think this is mainly about the content material, but rather about how the sentences and paragraphs are compiled. For example, section 3.2 could start with a short paragraph, providing a break-down of the steps involved in the architecture, before describing each part in following paragraphs. I recommend the authors to take a look at Gopen and Swan (1990) regarding how to write more clearly and effectively.

➔ To make the neural network architecture comprehensible, we added a more non technical description of the overall architecture. We also went through the entire manuscript again to clarify various items.

4. The issue of non-conservation of interpolation algorithms is a major concern in a physical application as presented here. The basic model equations are derived from principles of mass and energy conservation. Therefore, if there are conservation violations induced by this method, this aspect needs to be clearly brought forward throughout the manuscript, to make sure it is not overlooked by readers. This aspect can be mentioned in Sec. 3.4, brought up as part of the results, for example by comparing the kinetic energy and the velocity spectra from the fine-resolution and interpolated fields. The short discussion in L. 234 onwards might be more suitable in a discussion paragraph.

➔ We admit that mass conservation is a possible issue in interpolation. However, most available interpolation methods do not conserve mass and are not designed for that. The goal of this study was to compare NN against such interpolation methods. We therefore think that this topic is somewhat beyond the scope of the current paper. However, we admit that it would be of great value to develop a dynamically

constrained NN interpolation method that also ensures mass conservation. This would be a topic of future research.

5. The results need to be structured more clearly. Right now, the sequence of results and examples in Sec. 4.1 appear somewhat arbitrary. While there certainly is some reasoning behind the structure and examples, it is not spelled out clearly, and thus the reader is left guessing about how to "connect the dots". Coming from an atmospheric science perspective, I suggest a structure that starts one specific case as an example, such as one of the frontal bands shown in Fig. 3., where the linear interpolation has clear deficiencies. Thereby, it would be helpful to also show additional atmospheric variables to illustrate the case (for example surface pressure or air temperature. A tropical cyclone or a Rossby wave breaking could be other interesting situations to present. After stepping through the example case, more statistically robust information could be provided, from considering a larger number of days or cases. Finally, you proceed to the application with the trajectory calculations, before considering energy conservation.

➔ Thanks a lot for this remark. We went through the section one more time to better explain our results. In summary, our structure of the results is to show first the interpolation method and then use it in a simulation. In the present work we only consider the horizontal velocity fields. Thus, first we show that the new interpolation method using the neural network outperforms linear interpolation and then we use these interpolated fields to demonstrate that also the trajectory simulation using these interpolated fields is better. The frontal bands shown in Fig. 3 are a coincidence, since the plot shows the error, which is highest at these bands.

6. On many occasions, the results are presented in a qualitative way (closer/larger/etc.). In order to connect the results to the figures, and to make it possible to follow the interpretation and evaluation of the authors, it would be very useful to include concrete numbers alongside the qualitative interpretation, while referring to the respective figure panels. Examples are L. 182 and onward, L. 198 and onward.

➔ We follow the methodology in presenting our results in tables and figures in a quantitative manner with associated qualitative descriptions being provided in the text.

7. The writing style of in particular the results section should be more distanced or objective. Now, the authors frequently use expressions like "we demonstrate", "we show" in the start of a paragraph, i.e. before actually having presented the evidence. As a critical reader, one might get the impression you are overselling the results. I strongly recommend changing this unnecessary forceful writing style to a more distanced, objective style. Let the reader see the evidence for themselves, while guiding them through the material, before drawing conclusions. Many paragraphs in the results are currently "upside down" in that way.

➔ We appreciate your remark and went through the entire manuscript again to soften some of our writing style.

8. As another, related aspect, the figures are not properly described. At present, the length of the text describing the results is very much out of balance with the number of figures. For example in Sec. 4.1.1, L. 183, an entire 3 figures are referred to within just 3 sentences, but none of the sentences describes what actually is seen within the figures. Rather than leaving it up to the author to interpret the figures, use some sentences for each figure to describe what is displayed, and highlight what is important to take away. This applies to all figures in the manuscript.

➔ Thank you for pointing this out, we extended the description of the figures and softened some of our writing style.

9. On several occasions (including Fig. 2, 4, 5), the figure captions contain information about the method or results that are not mentioned in the text. Such information must be placed in the main text.

➔ Our style of writing the manuscript is based on explaining the content of the figures in the captions and when referencing the figures commenting the results observed. We also went through the entire manuscript again to clarify various items.

10. What are the limitations of the method in terms of computational effort? In L. 192, it is briefly mentioned that the computation time is a factor 10 larger than linear interpolation. Is there still an advantage of neural network approach compared to for example quadratic interpolation? This could be worth a short section in the discussion. The improved conservation of other properties is also interesting, but unfortunately not shown in more detail.

➔ At this stage it is too premature to compare the efficiency. We are also not implementing the method in the most efficient way. For a true comparison we would need to implement it in the best way also in FLEXPART and then compare the execution time.

**Detailed comments:**

Title: "deep learning inspired": unclear what this expression means, consider to remove/replace. State what aspect of trajectory calculations is improved (accuracy).

➔ We changed the title to *Improving trajectory calculations by FLEXPART 10.4+ using single image superresolution*.

L. 20: Can you back up this statement by a reference/example?

➔ We added a reference.

L. 21: "where a dense network" rephrase. If the point is that the numerical weather prediction process produces large amounts of gridded data, then it would be sufficient to state just that, without mentioning observations (which are not at all part of this manuscript). Remove "reanalysis model", a reanalysis is generated from regular NWP models.

➔ We simplified the sentence to say that NWP and observations generate large amounts of gridded data.

L. 25 onward: check citation of references, missing brackets.

➔ We added the missing brackets.

L. 27: remove "just to name a few"

➔ We removed it.

L. 30: logical gap, what is the connection to the previous paragraph?

➔ We removed the logical gap by moving the section "Related work" into the introduction.

L. 34: remove "surprisingly", this entirely depends on the perspective of the writer.

➔ We removed it.

L. 34: briefly define "convolutional neural network".

➔ We added that a CNN is a neural network whose layers are convolutions, which puts the input images through a set of convolutional filters, each of which activates certain features from the input.

L. 44: what do you mean by "variable-scale"?

➔ Here, variable-scale means that the neural network can cope with different resolutions of the wind fields. This way it can be applied multiple times to interpolate a meteorological field to the desired resolution. This is explained in the sentence after L 44.

L. 44: what to you mean by "deep" - how deep?

➔ Here, deep refers to the neural network having multiple layers.

L. 45: Rephrase: "showcase" sounds like snapshots or illustrations, but as a reader I look for reliable evidence.

➔ We rephrased "showcase" with "demonstrate".

Section 2: "Related work". This section does currently not serve a clear purpose, and is somewhat duplicate with the introduction. I recommend deleting this section here, and partly incorporating bits in the introduction, partly into a clearer method description.

➔ We moved the section "Related work" to the introduction.

Section 3: "Methods". This section would benefit from a first paragraph that explains your overall approach, followed by a section that discusses the choice of the neural network, based on the range of choices that exist, in an accessible writing style.

➔ We added the overall approach to the "Methods" section. The choice of the neural network is then described at the end of the "architecture" section.

Section 3.1: "Training data". The training data would be more natural to place after sections that describe the actual neural network and approach.

➔ We swapped the section "Training data" and "neural network architecture".

L. 82: Why could this seem little data? How much training is commonly needed?

➔ The phrase is misleading. We will just state the number of training files. It is difficult to say how much training data is needed, at least a few thousand samples.

L. 107: rephrase using more distanced and objective terms. It could provide depth to the study to present a less well-performing approach in an appendix.

➔ Indeed, a comparison of different models would be an interesting study. Here, however, the focus is on improving the trajectory simulation.

L. 114: remove "for testing purposes"

➔ We removed it.

L. 117: 50 or 88 -> 50 and 88

➔ We changed "or" to "and".

Figure 1: several abbreviations and terms of the operations in the figure are not defined, include in caption or describing text. What do the bracketing lines indicate? The hierarchy between (a), (b) and (c) and between (a) and (d) could be made clearer in the figure, e.g. by lines that indicate "zooming in".

➔ We added an explanation about the dotted line, which just means that the ResBlock is repeated multiple times. It is difficult to indicate "zooming in" by lines in the figure.

L. 123: Add a statement about the purpose of the error metric, i.e. what is to be assessed.

➜ The error metrics are evaluating the accuracy of the interpolation and trajectory simulation. We add a statement to the revised version.

L. 127: here and elsewhere: ground truth -> truth. (ground truth would only make sense in a remote sensing context)

➜ We replaced "ground truth" with "truth".

The notations for RMSE and SSIM could be simplified and clarified, for example using \hat{z} for the interpolated quantity, and using a,b instead of x,y (which is commonly used for spatial coordinates) for the two figures in the SSIM metric. How important is the "perceived similarity of two images" for the given application? This would be a suitable place to mention conservation issues due to interpolation.

➜ Indeed, for x and y we refer to two images, to avoid confusion we use now a and b. We include the SSIM metric because we interpret the gridded horizontal velocity fields as images.

L. 142, 144: unclear what "this" refers to.

➜ 142: Replaced '...this is outside of the scope..' with '...directly implementing the neural network interpolation into FLEXPART is outside...'
➜ 144: Replaced 'This does not make full use...' with 'Using gridded up-sampled testing data does not make full use..'

L. 145-149: unclear, please rephrase

➜ We are not sure what exactly was unclear. However, we replaced this text with the following one and hope it is clearer now: "Using gridded up-sampled testing data does not make full use of the neural network capabilities, since the neural network only produced values at a fixed resolution of $0.5^\circ\times0.5^\circ$ latitude/longitude, while we still use linear interpolation of the wind data to the exact particle position when computing their trajectories. However, the neural network could in principle also determine the wind components almost exactly at the particle positions upon repeatedly using the trained SISR model to increase the resolution high enough to obtain the wind values at the respective particle positions."

L. 152 onward: the emphasized names do not appear to be re-used in the remainder of the manuscript. Maybe rather introduce 3-letter abbreviations, such as REF, LIN, NNI that then can re-appear in the results and figures.

➜ We removed the emphasized names.

L. 160: place references at the end of sentence

➔ Unfortunately, the sentence will become confusing when the references are not placed after the first sub-sentence (before the comma), since the explanation of the equations follows.

L. 163: clarify whether Xn, Yn are vectors with m elements, or for a specific time along the trajectory

➔ We added the time variable to the equation for clarification.

L. 172: This section seems to describe your approach, and would be better placed in the methods.

➔ We added a paragraph to the methods section to better explain our approach. Nevertheless, to remind the reader of the approach we leave the explanation here, too.

Figure 2: the lines for lin u and lin v are exactly equal, is this coincidence? x-axis is lacking a unit. RMSE is defined with an index, but given without index here. Please explain in the result text how to interpret this figure.

➔ The linear interpolation is not dependent on the data in contrast to the neural network interpolation which is trained on different data. The interpolation of Fig. 2 is explained in L. 176 ff.

Table 1: What do the arrows indicate? The caption contains a key result, that should be moved to the main text.

➔ The arrows indicate that for the RMSE a lower number while for the SSIM a higher number refers to a better interpolation. The key result is spelled out in text in L. 190.

L. 183 to 188: Need to guide the reader through the results. The comparison needs more structure, and quantitative examples from the figure where available to support the qualitative conclusions.

➔ We will extend the interpretation.

L. 191: Maybe express in relative terms, hardware-dependent?

➔ We now state that the linear interpolation is about 10 times faster than the neural network interpolation considering our hardware.

L. 195: distinguish "evaluate" and "apply" - an objective way to present the results would be to apply the method, display the results, and thereafter evaluate based on the error metrics.

➔ We reformulated the sentences to present the results in an objective way.

Figure 3: Lacking panel labels. The top row does not seem to give additional information to the bottom row. I recommend using a continuous color scale; the two-color scale gives unjustified importance to errors larger than 5 m/s. Maps are missing coordinates. The RMSE in the title should be part of the text rather than a caption title. It would be useful to present a specific situation with meteorological fields for context.

➔ We split the figure in sub-figures with panel labels. The top row shows that high errors occur at fronts, this is then shown in a zoomed-in sub-figure in the bottom row. The color scale emphasizes the high errors, this way we see the strong difference in the error at the fronts.

L. 198: "before": rephrase

➔ Here, "before" referenced the previous section and we changed "before" to "one time up-scaling" to make it clear.

L. 199: "relative error reduction": where shown?

➔ The error reduction is shown in Table 2., we added a reference.

L. 200: "This holds...": can this information be presented as part of a more aggregated and thus robust result?

➔ First we showed an example and then using Table 2 the result is presented in a robust way.

L. 202: unclear, rephrase

➔ We now state that the neural network interpolation is 19% more accurate than the linear interpolation.

Figure 4: See comments about Fig. 3, the color scale gives unjustified emphasis to wind errors above 2.5 m/s. Indicate in Fig. 3 where this zoom is taken. Arrows are difficult to see, take to separate panels, and use meteorological fields (e.g. sea level pressure or potential temperature) as reference in both sets of panels.

➔ We split the figure into sub-figures. Also here we want to emphasize high errors. The Arrows for the neural network interpolation almost coincide with the truth, thus the arrows of the true field are difficult to see.

Figure 5: This figure needs more explanation. What bins have been used? With only 10 bins, it may be more appropriate to show the lines as step function. What error metric has

been used? Can this figure be constructed on more than just one day to make it more robust?

➔ For each pixel we compute the relative error against the truth and increase the count of the corresponding bin (using 20 bins now).We now compute the error frequencies for all 138 levels and over 24h. This way the result is more robust.

Figure 6: see Fig. 3.

Table 2: see Table 1

Figure 7: consider to remove this Figure. At this point, quantitative information may be sufficient/more useful than another illustration

➔ Quantitative information is given in Table 2. However, we consider it important to also show the error structure at a concrete example, and this is shown in Fig. 7. We combined the previous Fig. 3 and 4, and also Fig. 6 and 7.

L. 209: likely -> conceivable, provide reference

➔ Replaced the sentence with: 'However, this does not necessarily mean that trajectories advanced using the neural network interpolated fields are more accurate. Trajectories are not always equally sensitive to wind interpolation errors,...'

L. 215: it would be useful to briefly re-cap how these results are obtained. One case, several cases, specific region? How are trajectory errors distributed on a global map, do they mirror the interpolation errors?

➔ Since the trajectory errors result from interpolation, trajectory errors (for relatively short trajectory duration) are distributed quite similarly to the interpolation errors. For trajectories of longer duration (say, 10 days or longer), errors would be smeared out over larger areas, since initial errors are propagated along the trajectories. We do not think adding a figure showing the trajectory error distribution would provide meaningful additional information.
➔ We added 'Here we show the results of the horizontal transport deviation (Eq. \eqref{eq:ahtd}) and standard deviations of particles advanced for 48 hours, using FLEXPART, after being initially globally distributed.'

L. 221: "smaller" - should this be "larger"?

➔ We reformulated the sentence to avoid confusion.

L. 223: "directly corresponds" - is this a result, your interpretation, or an assumption?

➔ Replaced with: '...interpolated ones, is likely a result of the lower frequency...'

L. 228: These paragraphs would better fit into a discussion section, together with other limitations. If possible, it would be useful to give more details, such that other studies can refer to your work.

→ We feel that there is not enough material to justify a separate discussion section. In a nutshell, all existing interpolation methods, to the best of our knowledge, are not conservative and if conservation on the level of interpolation is important, then different design choices on the level of interpolation are necessary altogether not only for neural network but also for polynomial interpolation.

L. 242: remove "just to name a few"

→ We removed it.

L. 245: would be useful to connect to weather phenomena here

→ Indeed, it is helpful to connect to weather phenomena. Therefore, we have added a more detailed discussion on the interpolation errors along the cold front shown in Figure 4.

L. 250: remove "see Fig. 2"

→ We removed it.

L. 263: this is an important limitation and should be taken up at different locations in the manuscript, including a discussion section. If non-conservation is an issue here, it would be useful to quantify. This would also give some balance to the study, which now mainly focuses on the advantages.

**Reviewer 2**

**Major comment**

Construction of the degraded data: On line 85 it is described that the lower resolution data was obtained from sub-sampling the original ERA5 data. I am a bit surprised by this approach, since it does not necessarily reflect the representation in a coarser-resolution model, where the state variables in a larger grid cell should still represent the average in this grid cell and not a sub-sample. Could you please comment on the choice of this degradation strategy.

One direct result of the approach could be the large differences across frontal systems as indicated for the linear interpolation of coarse vs reference data. Likely, these differences would be smaller when average would have been used for degrading.

➔ The motivation for our choice of degrading data by leaving certain grid points entirely unchanged, is twofold:

3. This approach has been used often in the past when studying wind interpolation errors for trajectory models (e.g., Kuo et al., 1985; Stohl et al., 1995). It is appropriate to be consistent with such past approaches.

4. We agree that it would also be interesting to see how higher-resolution data could be reconstructed from lower-resolution data. This would then be more equivalent to downscaling approaches in weather prediction. However, for comparing the skill of different interpolation methods, this is not ideal. At the points where data are available at both high and low resolution, these data would be different in each case. Reconstruction of points in between would then not only reflect differences in the skill of interpolation but also the data differences at the points from where the interpolation is done. This would thus

not allow a "clean" evaluation of different interpolation methods, mixing the effects of interpolation and grid-cell averaging for the coarse-resolution data points.

**Minor comments**

L40: Higher-order interpolation. It would be interesting to see how higher-order interpolation schemes would compete with the ML approach. Did you give this any try?

➔ At this stage it is too premature to compare the efficiency. We are also not implementing the method in the most efficient way. For a true comparison we would need to implement it in the best way also in FLEXPART and then compare the execution time.

L65ff: Largely repeating the same points and references as in the introduction. Consider removing/shortening it here or in the intro.

➔ We moved the section "Related work" to the introduction.

L83: I would rather call this a 'vertical model layer' than a 'horizontal layer'.

➔ The input of the neural network is a horizontal u or v velocity component. Here, "vertical model layer" refers to the horizontal u or v velocity.

L87: How much does the exclusive treatment of the horizontal wind components impact the flow's mass budget (continuity)? It is mentioned later (conclusions) that all interpolation methods suffer from potentially breaking conservation laws and that physics-based ML could improve things. Maybe it can already be mentioned here. Why was the vertical wind not included in this study? Are there any fundamental differences that make it impossible to directly train the model for vertical wind?

➔ The vertical velocity is fundamentally different from the horizontal velocity as it is much more small scale. In practice the vertical velocity will require training of a more complicated neural network as neural networks have a tendency to learn large scale features first, which is referred to as a spectral bias. For this study we did not have the computational resources to experiment with the vertical velocity.

L115: Original levels are counted from the model top in IFS. So 0 to 50 would be the upper part of the atmosphere. What is the rational for cutting at level 50? What is the approximate pressure at this level? Does this separate into troposphere vs stratosphere?

➔ We have it bottom to top. Cutting at index 50 (ca 8000m) results in separating troposphere and stratosphere and above

Related to training two models for two vertical layers. How about training different models land and ocean as these give fundamentally different lower boundary conditions. How much does the performance increases in the ML method differ for land and ocean areas? How much for boundary layer (where turbulence is part of FLEXPARTs transport description) vs free troposphere?

➔ Indeed, distinguishing land and ocean in the training would be an alternative to our rather simple differentiation by height levels. However, there are also many other potential alternatives, such as developing different training data sets for climatically different regions (e.g., tropics, subtropics, midlatitudes), within or above the boundary layer, or for different meteorological situations. For developing an optimal method, it will be important to explore several of these options but it is beyond the scope of the current exploratory paper. With respect to the boundary layer, it is important to note that turbulence parameterizations have been switched off in FLEXPART for the current paper, as we wanted to study interpolation errors in isolation.

L131: Are mu_x and mu_y scalars representing the overall image mean? If yes, I don't quite understand the use of the 11 x 11 Gaussian filter. Furthermore, I think it would be good to argue if and why SSIM should be a useful metric for comparing wind components as opposed to images. I suppose wind components will have a very different pdf from that of images (color channels)?

➔ Here, we stated the definition of the SSIM as used in practise, which uses the 11x11 Gaussian filter. We agree that the SSIM is not a traditional error measure. However, since our model is an adaptation from an image processing task we felt it was reasonable to present also the SSIM measure as well as it is useful for the machine learning community.

L132f: What is the motivation for K1 and K2? Why not simply mention C1=1E-4 and C2=9E-4?

➔ For the sake of completeness we stated the definition of K1 and K2 as well.

L177: There is an exception to this observation! For SSIM linear interpolation in u seems to perform slightly better than model4.

➔ We changed it to almost always has better metrics

L186, Fig.5: How would the same figure look like for the relative error? Are these large error associated with large wind speeds?

➔ The largest errors generally occur where wind shears are largest, and this is usually associated with fronts and, generally, higher than average wind speeds. We discuss this now in more detail in the discussion of Fig. 4, which presents a clear example of this.

➔ We updated figure 5 to present the relative error.

L191: It is mentioned elsewhere that FLEXPART was not run on the same compute architecture as the ML model. How comparable are the times given here? Consider adding CPU/GPU specs.

➔ The training of the neural network is done on the mentioned GPU device. The FLEXPART simulations are run on a CPU. Since we updated the interpolated fields before the simulation, all simulation times are the same.

Fig 6: Figure caption wrong? I assume these are similar differences as in Fig. 3

➔ After reading the caption of Fig. 6 again we could not find an error.

Fig 7: Why do we not see the checkerboard pattern (as mentioned in the caption to Figure 4) here?

➔ We do see the checkerboard pattern, however it is only every fourth pixel that stays the same, this way the checkerboard pattern is less visible.

L234ff: Other downscaling approaches ingest additional high-resolution predictor variables (like topography or land cover) that have a direct impact on near-surface flow and spatial variability. Could such predictors be integrated into the present method as well?

➔ This is a very interesting idea and could be integrated into the current method. However, it would be another scope and thus relevant for future research.

**Technical issues**

Citation style: Seems to be wrong. Authors are given outside braces most of the time.

➔ Thanks for pointing this out, we correct this.

Equation 1: Consider using the same x, y notation as in equation 2.

➔ Indeed, for x and y we refer to two images, to avoid confusion we use now a and b.

L188: Additional figures in git repository? Shouldn't they rather be made available as part of a supplemental document/dataset? As git repository is not a permanent link/location, I would suggest to put figures elsewhere.

> ➔ The code and additional figures are now stored in a zenodo repository.

---

## Author Response (AR2)

**Summary**

We thank the reviewer for providing valuable feedback on the second version of our submitted paper. We have done our best to take into account all remarks raised. However, we disagree with the first comment, which we explain in detail below. In the following we give a detailed list of all the changes made in response to the points raised by the reviewer. Thank you once more for your help in improving our paper.

**Main comments**

**Comment 1**

Both reviewers have commented that the selection of every other grid point etc. to degrade the wind field is not consistent with what the information on the model grid represents. Even if the authors find studies that have used the same approach before, the problem remains that there is a spatial sub-sampling of the discretized fluid, rather than a corser representation in terms of averaged properties. The sub-sampling at an interval violates the perception of the fluid as a continuity, which matters for interpolation. Many other members of the targeted audience of geophysical models will have the same reaction as the two reviewers, and will immediately be sceptical to your method and results.

Therefore, it seems to me a moderate but necessary adjustment to your study to average 2x2 and 4x4 grid points to obtain the coarser version of their training and test grids.

Even if this is the first study of this kind in atmospheric science, it should get the community interested into a new methodology, rather than raise scepticism. I therefore strongly recomment that such a basic aspect of the fluid no be overlooked.

**Response 1:**

We believe that the reviewers confuse aspects of kinematics vs. dynamics of the flow with this point. In our study, we present a method for interpolating wind fields for a kinematic trajectory model (which always uses interpolated winds), whereas the reviewers focus on the dynamical consistency of the flow. We fully agree that, if the goal was to build a neural network that reconstructs a higher-resolution flow from a coarser-resolution one, then the correct approach is as suggested by the reviewers. This would also be possible as an alternative basis for trajectory calculations.

However, to assess the quality of the interpolation, we need a ground-truth state. Now, if we average the wind fields to a coarser resolution, we would also use information from the grid points that we actually want to reconstruct by interpolation. A comparison between high-resolution and low-resolution winds is then not meaningful anymore, since any differences are a combination of **averaging and interpolating**. By contrast, we want to assess the quality of the **interpolation only**. In addition, the interpolation distance to the high-resolution ground-truth grid points (averaging 2x2 grid points, with the new grid point in the centre of these) would then only be half a grid distance, and over such a short distance, interpolation errors will always be very small. So our test, in addition to mixing the effects of averaging and interpolating, would also become less rigorous.

This is also the reason why past studies have used the same technique as we have used, in order to compare different interpolation techniques vs. a ground truth. We see no good reason for deviating from that practice.

**Comment 2**

The argumentation about performance is somewhat contractictory or uneven. The introduction heavily emphasizes how surprising it is that the "simple" linear regression is still being used. However, this is by no means a surprise. Rather, previous studies have shown that the cost-benefit ratio of higher-order interpolation did not justify other interpolation methods. This is actually stated in L. 59.

Towards the outlook section, you provide an estimate of 1 order of magnitude increase in computation time from the single image superresolution approach to obtain 20-50% lower AHTD. These are quite the same numbers you cite for higher-order schemes in L. 59 (1 order of magnitude, 30%).

I do not find the conclusions balanced in light of these facts. There is quite some overhead with implementing GPU-enabled model code, training, etc. If the same gains can be achieved with simple higher-order interpolation, why is it worth exploring your methods further? I am sure the authors can come up with an answer to this question, but it would be nice to see this properly stated.

**Response 2:**

This is the first attempt to analyze the accuracy of neural network interpolation for Lagrangian trajectory computations. We choose to compare against the standard method of linear interpolation. For future work it could be interesting to compare our method against other interpolation methods such as higher order polynomial interpolation. At this stage we are also not implementing the method in the most efficient way (we generate the higher-resolution fields offline and then read them into FLEXPART). For a true comparison we would need to implement the neural network implementation directly into FLEXPART and then compare the execution time.

**Comment 3**

There are still numerous hard-to-read sections in the manuscript. I make some recommendations in the minor comments below. I recommend the authors read some instructions on how to improve the clarity of scientific writing (Gopen and Swan, 1990; Schultz 2009)

**Response 3:**

We appreciate your remark and went through the entire manuscript again to soften some of our writing style.

**Detailed comments**

L. 13: we demonstrate -> we find
- ➔ We changed it.

L. 34: rephrase in light of major comment #2
- ➔ We rephrased it.

L. 45-49: This paragraph very similar to L. 24, shorten/rephrase
- ➔ We shortened the paragraph.

L. 50, 53, 56: simple/surprising: rephase in light of major comment #2
- ➔ We rephrased surprising.

L. 70: project -> study
- ➔ We changed it.

L. 71: variety -> range. Please back-up statement with a reference. Maybe it would be more correct to state that this can be the case, but there are for example differences between small-scale turbulence and horizontal turbulence.
- ➔ We softened our statement: For the training and evaluation we consider that meteorological fields are characterized by self-similarity over a range of spatio-temporal scales. This means that the structure of the field from one resolution to a higher one is similar.

L. 97: most impressive: state objectively, e.g. results with smallest AHTD
- ➔ We changed it to:  the EDSR network giving the lowest interpolation error and having the shorter training time.

L. 97: greatest ease of training -> most straightforward training. It is not clear what this means in practice.
- ➔ We changed it to:  the EDSR network giving the lowest interpolation error and having the shorter training time.

L. 97: exclusively -> only
- ➔ We changed it.

L. 102: ...levels are counted... -> level indexes increase upward, contrary to ECMWF
- ➔ We changed it.

L. 106: this choice of method is not consistent with the concept of a continuous fluid, which matters for interpolation, see major comment #1.
- ➔ See answer to comment #1.

L. 120: how is the structure different, and how has this been quantified?
- ➔ When looking at the windfields we saw that the lower fields have more small scale structures compared to the higher levels (see below).

[Figure]

upper level

[Figure]

lower level

L. 176: compute trajectories: this should be part of the methods, rather than the results. Maybe does not need to be mentioned here.

➔ We removed it.

L. 177: we demonstrate -> we compare the accuracy

➔ We changed it

L. 180: we demonstrate -> we investigate

➔ We changed it.

L. 182: we demonstrate -> we proceed with

➔ We changed it.

L. 184: this sentence needs to be expanded to a full description of what is seen in Fig. 2. Deciphering the meaning of this figure can not be left to the reader.

➔ We added the explanation: Here, each neural network is used to interpolate each resolution, starting with the resolution the model is trained on.

L. 188: we consider -> we first consider

➔ We changed it.

L. 211-219: It was not possible for the reviewer to comprehend what is described here, a figure or table?

➔ We summarize the data from the table. (?)

Figure 2: the caption needs to be rephrased to describe panel contents. Methodological statements need to be moved to the main text.

➔ In the main text we now also give the methodological statement.

Figure 4: Methodological statements need to be moved to the main text.

➔ (?)

L. 229: this is indeed the case -> restate what is "this"

➔ We reformulated the sentence to: Fig. 6 shows that trajectories that are advanced using neural network interpolated wind fields closer to trajectories that are advanced using the original "ground-truth" wind fields compared to linear interpolated wind fields.

L. 243: we have also checked -> how has this been done
L. 245: slightly better: how has this been quantified?
➔ We changed the paragraph to:We have also checked how well the quasi-conserved meteorological property of potential vorticity is conserved along the trajectories by computing absolute and relative transport conservation errors along trajectories in the stratosphere. We selected particles that were not affected by convection or boundary layer turbulence by selecting trajectories within the stratosphere that never traveled through space where the relative humidity exceeded 90\%. A full explanation of the method we used can be found in [stohl1998].The absolute and relative transport conservation errors of potential vorticity showed insignificant differences between the different trajectory data sets.

L. 251: restate what "this" refers to
➔ We reformulated the sentence to: This way neural network interpolation could make semi-Lagrangian advection schemes much more accurate.

L. 285: see the papers -> see the studies
➔ We changed it.